# The Efficacy of Stretching Exercises on Arterial Stiffness in Middle-Aged and Older Adults: A Meta-Analysis of Randomized and Non-Randomized Controlled Trials

**DOI:** 10.3390/ijerph17165643

**Published:** 2020-08-05

**Authors:** Michitaka Kato, Fumi Nihei Green, Kazuki Hotta, Toshiya Tsukamoto, Yasunari Kurita, Akira Kubo, Hisato Takagi

**Affiliations:** 1Department of Physical Therapy, Shizuoka, Faculty of Health Science, Tokoha University, Shizuoka 420-0911, Japan; t-tsukamoto@sz.tokoha-u.ac.jp (T.T.); ykurita@sz.tokoha-u.ac.jp (Y.K.); 2Anti-aging Center, Ginza Hospital, Tokyo 104-0061, Japan; fng@medica21.com (F.N.G.); ak@kuboakira.com (A.K.); 3Department of Physical Therapy, Niigata University of Health and Welfare, Niigata 950-3198, Japan; kazuki-hotta@nuhw.ac.jp; 4Department of Cardiovascular Surgery, Shizuoka Medical Center, Shizuoka 411-8611, Japan; kfgth973@ybb.ne.jp

**Keywords:** stretching exercises, arterial stiffness, endothelial function, middle-aged, older adult

## Abstract

Background: Aerobic exercise is known to reduce arterial stiffness; however, high-intensity resistance exercise is associated with increased arterial stiffness. Stretching exercises are another exercise modality, and their effect on arterial stiffness remains unclear. The purpose of this study was to determine whether stretching exercises reduce arterial stiffness in middle-aged and older adults, performing the first meta-analysis of currently available studies. Methods: We searched the literature for randomized controlled trials (RCTs) and non-RCTs published up to January 2020 describing middle-aged and older adults who participated in a stretching intervention vs. controls without exercise training. The primary and secondary outcomes were changes in arterial stiffness and vascular endothelial function and hemodynamic status. Pooled mean differences (MDs) and standard MDs (SMDs) with 95% confidence intervals (CIs) between the intervention and control groups were calculated using a random effects model. Results: We identified 69 trials and, after an assessment of relevance, eight trials, including a combined total of 213 subjects, were analyzed. Muscle stretching exercises were shown to significantly reduce arterial stiffness and improve vascular endothelial function (SMD: −1.00, 95% CI: −1.57 to −0.44, *p* = 0.0004; SMD: 1.15, 95% CI: 0.26 to 2.03, *p* = 0.01, respectively). Resting heart rate (HR) and diastolic blood pressure (DBP) decreased significantly after stretching exercise intervention (MD: −0.95 beats/min, 95% CI: −1.67 to −0.23 beats/min, *p* = 0.009; MD: −2.72 mm Hg, 95% CI: −4.01 to −1.43 mm Hg, *p* < 0.0001, respectively) Conclusions: Our analyses suggest that stretching exercises reduce arterial stiffness, HR, and DBP, and improve vascular endothelial function in middle-aged and older adults.

## 1. Introduction

Vascular aging results in stiffer arteries and vascular endothelial dysfunction, and may have a role in the development of cardiovascular disease [1]. In particular, excess reactive oxygen species production by mitochondria is a key mechanism of aging and age-related vascular dysfunction [2]. Physical inactivity is an independent risk factor for the deterioration of vascular function, atherosclerosis, and cardiovascular diseases [3]. Regular exercise training leads to the prevention of cardiovascular diseases and mortality. Aerobic exercise is known to significantly reduce large artery compliance, one of the parameters of arterial stiffness, in middle-aged and older humans [4]. On the other hand, a previous meta-analysis has shown that resistance exercise does not decrease arterial stiffness in middle-aged subjects [5].

Stretching exercises are another exercise modality along with aerobic and resistance exercises. Stretching exercise has been widely utilized as a warm-up prior to main exercise sessions [6]. A study reported that stretching exercises reduced arterial stiffness and improved endothelial function in middle-aged women [7]. Although the results of a few studies have been published to date, no meta-analysis has been conducted, and it remains unclear whether stretching exercises reduce arterial stiffness in middle-aged and older adults.

The purpose of this study was to determine whether stretching exercises improve select markers of vascular function, including endothelial function and arterial stiffness in middle-aged and older adults, performing the first meta-analysis of currently available studies.

## 2. Methods

Studies evaluating the effect of stretching exercises on arterial stiffness, endothelial function, and/or hemodynamic status were searched through databases (MEDLINE and EMBASE) using web-based search engines (PubMed and OVID). We started to search the databases on 10th January 2020. In addition, relevant studies were manually identified from the references of initially identified articles and relevant reviews and commentaries. We used the following search terms: stretching, stretching exercises, stretching training, stretching intervention, arterial stiffness, vascular stiffness, pulse wave velocity, pulse wave analysis, endothelial function, and hemodynamic status. The search was limited to human studies in English. Furthermore, conference proceedings were also searched manually in an attempt to identify relevant unpublished studies.

There was no ethical approval because this study did not include confidential personal data and did not involve patient intervention.

Studies meeting the following criteria were included: the designs were randomized controlled trials (RCTs) or non-randomized controlled trials (non-RCTs); the study populations were middle-aged and older adults; exercise intervention groups were allocated to perform stretching exercise training as a physical exercise therapy with no other combined training; the control groups received instruction to continue their regular lifestyle habits; the durations of interventions were ≥4 weeks; outcomes included arterial stiffness, endothelial function, or hemodynamic status. We defined middle-aged and older adults as subjects whose age was 40 years or older. Three reviewers (M.K., Y.K., and T.T.) each reviewed all the eligible trials and determined whether they fulfilled the selection criteria. Disagreements were resolved by discussion.

The following data were extracted from each report: study design, the number of subjects, baseline participant characteristics including age, gender, and body mass index (BMI), and details of the stretching exercise intervention (stretched muscle, time per session, frequency, duration of intervention, and intensity). Data were extracted in duplicate by two investigators (A.K. and M.K.) and verified independently by a third (H.T.).

The primary outcome was changes in arterial stiffness. Brachial–ankle pulse wave velocity (baPWV), cardio-ankle vascular index (CAVI), or augmentation index (AIx) was used as a parameter of arterial stiffness [8,9,10]. The secondary outcomes were changes in endothelial function, hemodynamic status, and muscle flexibility. The reactive hyperemia peripheral arterial tonometry (RH-PAT) index or flow-mediated dilation (FMD) was used as a parameter of endothelial function, while heart rate (HR) and blood pressure were used as parameters of hemodynamic status [6,11]. A sit-and-reach test was used for measuring muscle flexibility.

The risk of bias for each study was assessed by two investigators (M.K. and H.T.) using the risk of bias tool in the Cochrane Handbook for Systematic Reviews of Interventions for RCTs and the Risk of Bias Assessment tool for Non-randomized Studies (RoBANS) for non-RCTs [12,13].

Continuous outcome measures were expressed as a change in the mean ± standard deviation (SD) from baseline to follow-up and were pooled as the mean difference (MD) or standardized mean difference (SMD) with a 95% confidence interval (CI). The SMD is used as a summary statistic in meta-analyses when the studies all assess the same outcome, but it is measured in a variety of ways (for example, if the primary outcome is arterial stiffness, studies use different parameters such as baPWV, CAVI, or AIx). Statistical heterogeneity was evaluated according to the Higgins *I^2^* statistic. *I^2^* values of 0 to 24.9%, 25% to 49.9%, 50% to 74.9%, and 75% to 100% were considered no, low, moderate, and high statistical heterogeneity, respectively [14]. A sensitivity analysis was performed to assess the contribution of only RCTs to the pooled estimate by excluding non-RCTs and recalculating the pooled SMD estimates for the remaining studies. In addition, other sensitivity analyses were performed as needed to enhance the results. Funnel plots were used to examine the presence of publication bias. A *p*-value of less than 0.05 was considered statistically significant. Analyses were carried out using Review Manager (version 5.3; The Cochrane Collaboration, London, UK).

## 3. Results

According to the search strategies, a total of 702 records were identified. Studies that did not meet the inclusion criteria, duplicate studies, studies with no reported change in outcome, and studies for which the full text was unavailable were excluded. From 69 potentially relevant citations retrieved from electronic database searches and manual searches of the reference lists, six RCTs [7,15,16,17,18,19] and two non-RCTs [20,21] fulfilled the inclusion criteria (Figure 1). The included studies had a total of 213 participants. All the studies included were designed to compare stretching exercises without combined training to the continuation of the subjects’ regular lifestyle habits (the control).

The baseline patient characteristics of the included studies are presented in Table 1. The sample sizes, mean age, and mean BMI ranged from 13–50, 44–71 years old, and 21.2–34.1 kg/m^2^, respectively. The proportion of females ranged from 0 to 100%. One of the eight studies included subjects with stable chronic heart failure [18], one included subjects with stable peripheral artery disease [19], and one included subjects with metabolic syndrome and chronic diseases [21]. The remaining five studies included subjects without cardiovascular disease or lifestyle-related diseases.

Table 1 also shows the details of the stretching exercise interventions of the included studies. The training types were an active static stretching intervention for major muscle groups in six studies, passive static intervention using a splint in one study, and combined intervention in one study. The training intensities were moderate (11 to 15 on the rating of perceived exertion (RPE) scale; the point of minimal discomfort, without pain) in six studies, and high (the point of maximal exertion defined as RPE > 18) in two studies. Training session duration, frequency, and the duration of the exercise training intervention ranged from 15−60 min/session, 3−7 times/week, and 4−12 weeks, respectively.

The risk of bias is summarized in Table 2. Although the subjects of all eight studies were not blinded, we considered that arterial stiffness, the primary outcome in this analysis, was not likely to be influenced by the lack of blinding. Therefore, we judged that the risk of bias related to the “blinding of participants and personnel” and “blinding of outcome assessment” was low.

Five studies evaluated the primary outcome, the effect of stretching exercises on arterial stiffness, and were included in the analysis (Table 3). One study assessed two different measurements of arterial stiffness, baPWV and AIx [15]. These values were extracted separately as Wong-1 and Wong-2. Similarly, another study measured baPWV and CAVI, which are both measurements of arterial stiffness [20]. These values were also extracted separately as Nishiwaki-1 and Nishiwaki-2. Primary analysis for arterial stiffness was conducted with five studies. For two studies that used more than one measurement for arterial stiffness, we used the data for baPWV because baPWV is the standard assessment for arterial stiffness. Four of the five studies in the primary analysis used baPWV and one study used CAVI. Arterial stiffness was significantly reduced by stretching exercises compared with the control (SMD: −1.00, 95% CI: −1.57 to −0.44, *p* = 0.0004, I_2_: 52%, Figure 2).

As Nishiwaki’s and Kim’s studies were non-RCTs, sensitivity analysis-1 was performed excluding these two studies. Sensitivity analysis-1 with only RCTs also showed that stretching exercises significantly reduced arterial stiffness compared with the control (SMD: −0.90, 95% CI: −1.63 to −0.16, *p* = 0.02, I_2_: 63%; Figure 3). 

Sensitivity analysis-2 was conducted using the non-baPWV measurements for the two studies with more than one arterial stiffness measurement. Five studies were included in sensitivity analysis-2 with two studies using baPWV, two studies using CAVI, and one study using AIx. Sensitivity analysis-2 also showed that stretching exercises significantly reduced arterial stiffness compared with the control (SMD: −1.19, 95% CI: −1.81 to −0.58, *p* = 0.0001, I_2_: 58%; Figure 4).

We performed sensitivity analysis-3 with only studies using baPWV. Sensitivity analysis-3 showed that stretching exercises also significantly reduced baPWV compared with the control (MD: −77.50 cm/sec, 95% CI: −135.48 to −19.53 cm/sec, *p* = 0.009, I_2_: 94%; Figure 5).

Finally, we performed sensitivity analysis-4 with only female subjects to explore the potential gender influence. Arterial stiffness was significantly reduced by stretching exercises compared with the control among female subjects (SMD: −0.86, 95% CI: −1.41 to −0.31, *p* = 0.002, I2: 47%; Figure 6).

Regarding the effect of stretching exercises on endothelial function, one of the secondary outcomes, three studies were included in the analysis (Figure 7, Table 3). Two studies evaluated the RH-PAT index and one study evaluated FMD. Stretching exercises were shown to significantly improve endothelial function compared with the control (SMD: 1.15, 95% CI: 0.26 to 2.03, *p* = 0.01, I_2_: 68%).

To evaluate the effect of stretching exercises on hemodynamic status, another secondary outcome, six studies were included in the analyses (Figure 8, Table 4). Stretching exercises significantly decreased HR and diastolic blood pressure (DBP) compared with the control (MD: −0.95 beats/min, 95% CI: −1.67 to −0.23 beats/min, *p* = 0.009, I_2_: 0%; MD: −2.72 mm Hg, 95% CI: −4.01 to −1.43 mm Hg, *p* < 0.0001, I_2_: 0%, respectively). There was no significant difference in systolic blood pressure between the exercise and the control groups.

To evaluate the effect of stretching exercises on muscle flexibility, four studies were included in the analysis. Stretching exercises significantly increased muscle flexibility compared with the control (MD: 7.18 cm, 95% CI: 1.33 to 13.02 cm, *p* = 0.02, I_2_: 87%; Figure 9).

Funnel plots of the primary outcome generated were symmetric, indicating that the results of those meta-analyses were not influenced by publication bias.

## 4. Discussion

The results of the present meta-analysis demonstrated that stretching exercises reduced arterial stiffness and improved endothelial function compared with the control. Furthermore, stretching exercises decreased resting HR and DBP. These data are derived from a meta-analysis of six RCTs and two non-RCTs enrolling a total 213 subjects. To our knowledge, this is the first meta-analysis to assess the potential benefits of stretching exercise on arterial stiffness.

The present study showed that stretching exercises improved endothelial function. Increased cardiovascular disease in aging is partly a consequence of the vascular endothelial cell senescence and associated vascular dysfunction [22]. Exercise training modifies blood flow, luminal shear stress, and tangential wall stress, all of which can improve endothelial function [23]. A recent meta-analysis demonstrated that aerobic and combined aerobic and resistance exercise increased FMD by 1.21% and 2.49%, respectively [11]. We confirmed that stretching exercises also improved endothelial function assessed by the RH-PAT index and FMD in our meta-analysis. A single session of stretching exercises improved endothelial function and peripheral circulation in patients with acute myocardial infarction. This improvement was potentially induced by increased NO production from vascular endothelial cells [6]. Cyclic stretch stimuli have been shown to upregulate the expression of endothelial nitric oxide synthase (eNOS) messenger ribonucleic acid in vascular endothelial cells in vitro [24]. Hotta et al. reported that 4 weeks of daily muscle stretching enhanced the endothelium-dependent vasodilatation of resistance arterioles in the skeletal muscle of aged rats [25]. In addition, in vitro studies showed that stretch stimuli on skeletal muscle cells increase intracellular superoxide dismutase (SOD) in skeletal muscle cells [26], resulting in decreased ROS generation. Increased eNOS and decreased ROS lead to the improvement of endothelial function [27]. Therefore, regular stretching exercises can improve endothelial function by increasing eNOS and intracellular antioxidants in skeletal muscle cells. Early atherosclerosis-related changes involve the impairment of endothelial function as an important initial step in the atherosclerotic process [28]. The impairment of endothelial dysfunction is a natural process with aging, and although its mechanism is largely unknown, a recent study shows that it may be related to the availability of epoxyeicosatrienoic acids and NO [29]. The change in endothelial function by stretching exercises may be helpful to prevent the progression of atherosclerosis.

The measurement of arterial stiffness is the most common non-invasive examination method for the detection of atherosclerosis-related changes [8]. Our meta-analysis demonstrated that stretching exercises reduced arterial stiffness with not only primary analysis but also sensitivity analyses. An improvement in arterial stiffness has been reported in previous studies immediately after performing short-term static stretching [30,31]. Furthermore, yoga practice, including muscle stretching, is reported to be effective in preventing or reducing arterial stiffness in elderly hypertensive patients [32]. These studies are consistent with our results, confirming the impact of stretching exercises on arterial stiffness. According to a previous meta-analysis, aerobic exercise had a significant effect on reducing baPWV (MD: −67 cm/sec, 95% CI: −97 to −38 cm/sec) [33]. In our meta-analysis, baPWV was significantly reduced after stretching exercises compared with the control (MD: −77.50 cm/m, 95% CI: −135.48 to −19.53 cm/sec). These results suggest that stretching exercises might lead to a comparable reduction in arterial stiffness compared with aerobic exercises. Yamato et al. reported that the reduction in baPWV following stretching exercises may have been the result of a change in peripheral arterial stiffness rather than a change in central arterial stiffness, which was the case with aerobic exercises [30]. The responses of arterial stiffness were different between aerobic and stretching exercises because aerobic exercise affects all arteries via the systemic circulation, whereas stretching exercise stimulates only vasculature in the muscle mechanically [30]. Although the mechanisms of the PWV decrease is not clear, it might be associated with the improvement of endothelial function following stretching exercises. There is a strong relationship between endothelial function and the arterial stiffness index in patients with atherosclerosis. In addition, arterial stiffness is functionally determined by the vascular tone of the artery, which is partially regulated by sympathetic nerve activity [4]. Eight weeks of stretching exercise has been reported to improve cardiac autonomic modulation by decreasing sympathetic activity and increasing parasympathetic nerve tone [16]. Therefore, decreased sympathetic nervous activation induced by stretching exercises could be one of the mechanisms underlying the reduction of arterial stiffness. Arterial stiffness is reported to be significantly and positively correlated with intima–media thickness in middle-aged subjects with and without diabetes [34], indicating that hyperplastic and/or hypertrophic change in vascular smooth muscle cells may have an impact on arterial stiffness. The effect of muscle stretching on vascular wall thickness remains unknown; however, one in vitro study reported the stretch-induced hypertrophic change of vascular smooth muscle cells [35]. It remains unknown whether muscle stretching causes such vascular morphological changes; however, at least, skeletal muscle stretching does not seem to have any unfavorable effects on arterial stiffness.

The present meta-analysis also demonstrated that stretching exercises decreased resting HR and DBP after the intervention period. Similarly, a study by Logan et al. showed that HR and DBP decreased by 3.31 bpm (*t* = 2.17, *p* = 0.05) and 2.13 mm Hg (*t* = 1.93, *p* = 0.07), respectively, following 20-min stretching exercises in healthy pregnant women, though they did not reach statistical significance [36]. HR and BP are controlled by the two branches of the autonomic nervous system, the sympathetic nervous system and parasympathetic nervous system. Regular exercise, like endurance training and yoga, is well known to enhance parasympathetic output [37]. As we described before, stretching exercises change cardiac autonomic modulation by increasing parasympathetic nerve tone [16]. Muscle stretching exercises are thought to be effective for regulating hemodynamics associated with changes in autonomic nerves.

The present meta-analyses have several limitations. First, the numbers of included studies and patients were relatively small, with only eight studies and 213 subjects, which may not be sufficient for a valuable meta-analysis. In future meta-analyses, it is important to assess more trials and subjects to confirm the effect of stretching exercises on arterial stiffness. Second, two of the eight studies in our analyses were non-RCTs. We included these two non-RCTs in the primary analysis because only few studies were available in this area. We also conducted a sensitivity analysis with only RCTs (sensitivity analysis-1), which showed similar results with the primary analysis. Third, the evaluation methods for the primary outcome, arterial stiffness, were not uniform, including baPWV, CAVI, and AIx. As supplementary analyses, we performed several sensitivity analyses to enhance our results (sensitivity analysis-2, -3, and -4). These results were all similar to the result of the primary analysis. Fourth, we were unable to evaluate whether stretching exercise is superior to aerobic exercises in reducing arterial stiffness. As we stated in the discussion, different stimulations from aerobic and stretching exercises may have different effects on arterial stiffness. Fifth, it is unclear whether stretching exercises have long-term effects on arterial stiffness in this meta-analysis because the durations of the included studies were mostly 4–8 weeks. Shinno et al. reported that stretching exercises decreased baPWV after intervention, though a significant increase in this value was observed after 6 months of not training. Therefore, stretching exercise might not have long-term effects on arterial stiffness once it is discontinued. Finally, the present study focused on the effect of stretching exercises on arterial stiffness but did not evaluate whether this effect leads to the prevention of diseases related to arteriosclerosis. Therefore, further studies will be needed to compare the effects of stretching exercises on disease prevention.

## 5. Conclusions

Our meta-analyses demonstrated that stretching exercises reduced arterial stiffness, HR, and DBP, and improved endothelial function, which are crucial parameters of arteriosclerosis in middle-aged and older adults.

## Figures and Tables

**Figure 1 ijerph-17-05643-f001:**
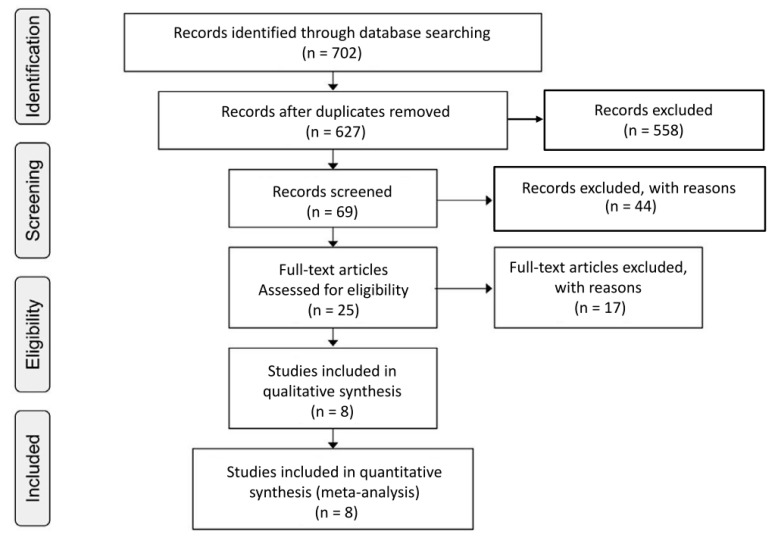
Flow chart of the systematic literature search for the meta-analysis.

**Figure 2 ijerph-17-05643-f002:**
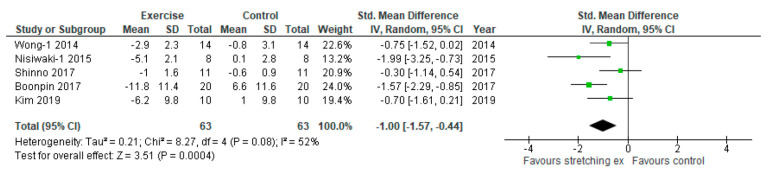
Forest plot comparing changes in arterial stiffness following stretching exercises to non-exercise controls (primary analysis for primary outcome). SD, standard deviation; IV, inverse variance; CI, confidence interval; Std, standardized.

**Figure 3 ijerph-17-05643-f003:**
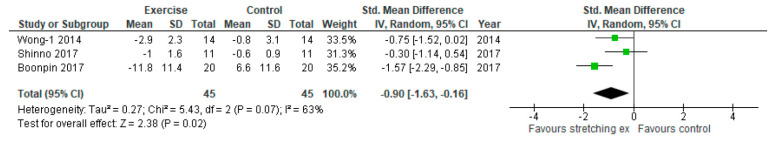
Sensitivity analysis with only RCTs comparing changes in arterial stiffness following stretching exercises to non-exercise controls (sensitivity analysis-1 for primary outcome). SD, standard deviation; IV, inverse variance; CI, confidence interval; Std, standardized.

**Figure 4 ijerph-17-05643-f004:**
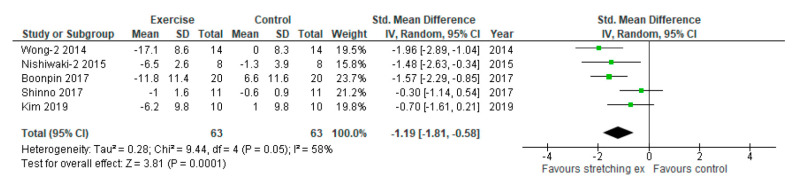
Sensitivity analysis with different assessments of outcome comparing changes in arterial stiffness following stretching exercises to non-exercise controls (sensitivity analysis-2 for primary outcome). SD, standard deviation; IV, inverse variance; CI, confidence interval; Std, standardized.

**Figure 5 ijerph-17-05643-f005:**
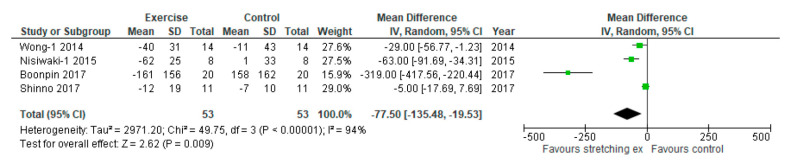
Sensitivity analysis with only studies using baPWV comparing changes in arterial stiffness following stretching exercises to non-exercise controls (sensitivity analysis-3 for primary outcome). IV, inverse variance; CI, confidence interval; Std, standardized.

**Figure 6 ijerph-17-05643-f006:**
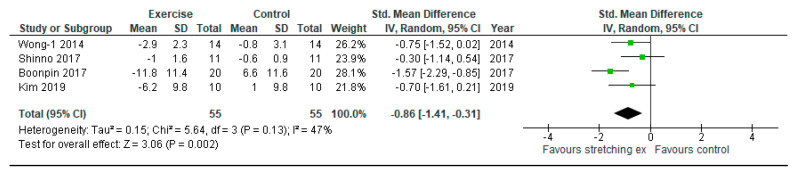
Sensitivity analysis-4 with only female subjects comparing changes in arterial stiffness following stretching exercises to non-exercise controls (sensitivity analysis-4 for primary outcome). IV, inverse variance; CI, confidence interval; Std, standardized.

**Figure 7 ijerph-17-05643-f007:**
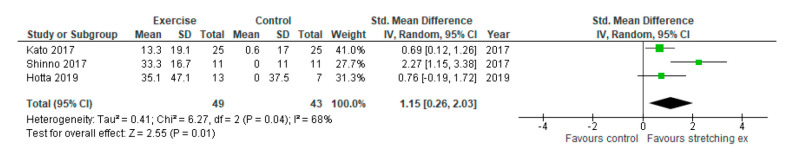
Forest plot comparing changes in endothelial function following stretching exercises to non-exercise controls. SD, standard deviation; IV, inverse variance; CI, confidence interval; Std, standardized.

**Figure 8 ijerph-17-05643-f008:**
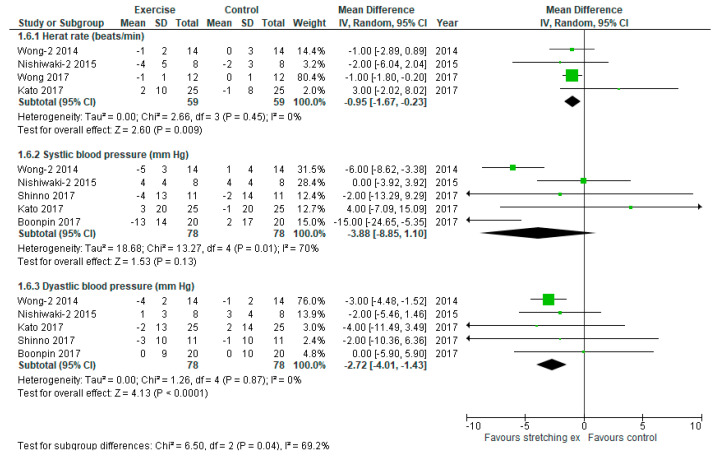
Forest plot comparing changes in hemodynamics following stretching exercises to non-exercise controls. SD, standard deviation; IV, inverse variance; CI, confidence interval.

**Figure 9 ijerph-17-05643-f009:**
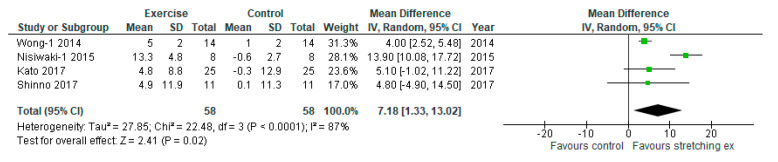
Forest plot comparing changes in muscle flexibility following stretching exercises to non-exercise controls. SD, standard deviation; IV, inverse variance; CI, confidence interval.

**Table 1 ijerph-17-05643-t001:** Characteristics of included studies.

Author	Design	Sample Size	Subject Characteristics				Stretching Exercise Intervention					Assessment of Outcome	
		Total	Age (years)	Female (%)	BMI (kg/m^2^)	Others (e.g., Complications)	The Stretched Muscle	Time per Session (min/session)	Frequency (times/wk)	Duration of Intervention (wks)	Methods and Intensity	Arterial Stiffness	Endothelial Dysfunction
Wong, 2014	RCT	28	57 ± 1	100	34.1 ± 1.1	Postmenopausal women	Active stretching training including upper and lower extremity muscles	50	3	8	Hold for 30 s at the point of maximal exertion	baPWV AIx	NR
Nishiwaki, 2015	Non-RCT	16	44 ± 4	0	23.3 ± 0.9	Healthy men	Active stretching training including upper and lower extremity muscles	30	5	4	Hold for 20 s at the point of minimum discomfort	baPWV CAVI	NR
Shinno, 2017	RCT	22	47 ± 3	100	21.2 ± 4.3	Healthy premenopausal women	Active stretching training including upper and lower extremity muscles	15	7	12	Hold for 20–30 s at the end range without pain	baPWV	RH-PAT index
Wong, 2017	RCT	24	58 ± 1	100	33.9 ± 1.3	Postmenopausal women	Active stretching training including upper and lower extremity muscles	50	3	8	Hold for 30 s at the point of maximal exertion	NR	NR
Boonpin, 2017	RCT	40	55 ± 3	100	24.8 ± 3.0	Postmenopausal women	Active stretching training including upper and lower extremity muscles	30	5	6	Hold for 20 s at the point of minimum discomfort	baPWV	NR
Kato, 2017	RCT	50	70 ± 9	22	22.7 ± 3.4	Stable CHF patients	Active stretching training including upper and lower extremity muscles	20	7	4	Hold for 30 s without pain	NR	RH-PAT index
Hotta, 2019	RCT with crossover	13	71 ± 2	46	33 ± 3.3	Stable symptomatic PAD patients	Passive stretching training using splint including lower extremity muscles	30	5	4	Ankle joints were kept at 15° of dorsiflexion without pain.	NR	FMD
Kim, 2019	Non-RCT	20	70 ± 2	100	26.7 ± 3.8	Metabolic syndrome and chronic disease women	Active and passive (combined) stretching training including upper and lower extremity muscles	60	3	8	RPE <= 11	CAVI	NR

RCT, randomized controlled trial; BMI, body mass index; wk, week; s, second; baPWV, brachial–ankle pulse wave velocity; AIx, augmentation index; NR, not reported; CAVI, cardio-ankle vascular index; RH-PAT, reactive hyperemia peripheral arterial tonometry; CHF, chronic heart failure; PAD, peripheral artery disease; FMD; flow-mediated dilation; RPE, rating of perceived exertion. Data are shown as mean ± SD.

**Table 2 ijerph-17-05643-t002:** Risk of bias.

**The Cochrane Collaboration: Tool for Randomized Studies**						
Studies	Random sequence generation	Allocation concealment	Blinding of participants and personnel	Blinding of outcome assessment	Incomplete outcome data	Selective reporting
Wong, 2014	Low	Unclear	Low	Low	Low	Low
Shinno, 2017	Unclear	Unclear	Low	Low	Unclear	Unclear
Wong, 2017	Low	Unclear	Low	Low	Low	Low
Boonpin, 2017	Unclear	Unclear	Low	Low	Unclear	Unclear
Kato, 2017	Low	Unclear	Low	Low	Low	Unclear
Hotta, 2019	Unclear	Unclear	Low	Low	Low	Unclear
**The Risk of Bias Assessment Tool for Non-randomized Studies (RoBANS)**						
Studies	Selection of participants	Confounding variables	Measurement of exposure	Blinding of outcome assessment	Incomplete outcome data	Selective outcome reporting
Nishiwaki, 2015	Low	Low	Low	Low	Low	Unclear
Kim, 2019	Low	Unclear	Low	Low	Low	Unclear

Low, low risk; High, high risk.

**Table 3 ijerph-17-05643-t003:** Changes in parameters of arterial stiffness and vascular endothelial function.

Author	Arterial Stiffness							Vascular Endothelial Function						
	Type	Baseline		Follow-Up		Absolute Change		Type	Baseline		Follow-Up		Absolute Change	
		SE	CON	SE	CON	SE	CON		SE	CON	SE	CON	SE	CON
Wong-1, 2014	baPWV (cm/sec)	1359 ± 29	1397 ± 40	1319 ± 33	1386 ± 47	−40 ± 31	−11 ± 43	NR	NR	NR	NR	NR	NR	NR
Wong-2, 2014	AIx (%)	35 ± 3	32 ± 2	29 ± 3	32 ± 3	−6 ± 3	0 ± 3							
Nishiwaki-1, 2015	baPWV (cm/sec)	1207 ± 28	1204 ± 25	1145 ± 19	1205 ± 38	-62 ± 25	1 ± 33	NR	NR	NR	NR	NR	NR	NR
Nishiwaki-2, 2015	CAVI	7.7 ± 0.2	7.6 ± 0.3	7.2 ± 0.2	7.5 ± 0.3	−0.5 ± 0.2	−0.1 ± 0.3							
Shinno, 2017	baPWV (cm/sec)	1158 ± 18	1162 ± 10	1146 ± 19	1155 ± 10	−12 ± 19	−7 ± 10	RH-PAT index	1.5 ± 0.25	1.6 ± 0.18	2.0 ± 0.25	1.6 ± 0.17	0.5 ± 0.25	0.0 ± 0.18
Wong, 2017	NR	NR	NR	NR	NR	NR	NR	NR	NR	NR	NR	NR	NR	NR
Boonpin, 2017	baPWV (cm/sec)	1364 ± 66	1390 ± 165	1203 ± 178	1482 ± 158	-161 ± 156	158 ± 162	NR	NR	NR	NR	NR	NR	NR
Kato, 2017	NR	NR	NR	NR	NR	NR	NR	RH-PAT index	1.81 ± 0.33	1.80 ± 0.22	2.05 ± 0.36	1.81 ± 0.35	0.24 ± 0.35	0.01 ± 0.31
Hotta, 2019	NR	NR	NR	NR	NR	NR	NR	FMD (%)	3.7 ± 1.2	3.7 ± 0.9	5.0 ± 2.0	3.7 ± 1.6	1.3 ± 1.74	0.0 ± 1.39
Kim, 2019	CAVI	9.7 ± 0.9	9.7 ± 0.9	9.1 ± 1.0	9.8 ± 1.0	−0.6 ± 1.0	0.1 ± 1.0	NR	NR	NR	NR	NR	NR	NR

SE, stretching exercises; CON, control; baPWV, brachial–ankle pulse wave velocity; AIx, augmentation index; CAVI, cardio-ankle vascular index; sec, second; RH-PAT, reactive hyperemia peripheral arterial tonometry; FMD; flow-mediated dilation; NR, not reported. Data are shown as mean ± standard deviation.

**Table 4 ijerph-17-05643-t004:** Changes in parameters of hemodynamic status.

Author	HR (beats/minute)						SBP (mm Hg)						DBP (mm Hg)					
	Baseline		Follow-Up		Absolute Change		Baseline		Follow-Up		Absolute Change		Baseline		Follow-Up		Absolute Change	
	SE	CON	SE	CON	SE	CON	SE	CON	SE	CON	SE	CON	SE	CON	SE	CON	SE	CON
Wong, 2014	64 ± 2	67 ± 3	63 ± 2	67 ± 3	−1 ± 2	0 ± 3	133 ± 3	137 ± 4	128 ± 3	138 ± 4	−5 ± 3	1 ± 4	77 ± 2	80 ± 2	73 ± 2	79 ± 2	−4 ± 2	−1 ± 2
Nishiwaki, 2015	66 ± 6	61 ± 2	62 ± 3	59 ± 3	−4 ± 5	−2 ± 3	119 ± 4	129 ± 5	123 ± 3	133 ± 3	4 ± 4	4 ± 4	78 ± 3	84 ± 4	79 ± 2	87 ± 4	1 ± 3	3 ± 4
Shinno, 2017	NR	NR	NR	NR	NR	NR	112 ± 14	113 ± 13	108 ± 12	111 ± 14	−4 ± 13	−2 ± 14	71 ± 10	68 ± 10	68 ± 10	67 ± 10	−3 ± 10	−1 ± 10
Wong, 2017	65 ± 1	66 ± 1	64 ± 1	66 ± 1	−1 ± 1	0 ± 1	NR	NR	NR	NR	NR	NR	NR	NR	NR	NR	NR	NR
Boonpin, 2017	NR	NR	NR	NR	NR	NR	137 ± 16	139 ± 19	124 ± 12	141 ± 13	−13 ± 14	2 ± 17	69 ± 8	71 ± 9	69 ± 10	71 ± 11	0 ± 9	0 ± 10
Kato, 2017	63 ± 9	64 ± 7	66 ± 11	63 ± 8	3 ± 10	−1 ± 8	125 ± 18	127 ± 19	128 ± 21	126 ± 20	3 ± 20	−1 ± 20	69 ± 13	71 ± 14	67 ± 13	73 ± 13	−2 ± 13	2 ± 14
Hotta, 2019	NR	NR	NR	NR	NR	NR	NR	NR	NR	NR	NR	NR	NR	NR	NR	NR	NR	NR
Kim, 2019	NR	NR	NR	NR	NR	NR	NR	NR	NR	NR	NR	NR	NR	NR	NR	NR	NR	NR

HR, heart rate; SBP, systolic blood pressure; DBP, diastolic blood pressure; SE, stretching exercises; CON, control; NR, not reported. Data are shown as mean ± SD. Generated funnel plots of the primary outcome were symmetric, indicating that the results of those meta-analyses were not influenced by publication bias.

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
