# Peer review of "The Efficacy of Stretching Exercises on Arterial Stiffness in Middle-Aged and Older Adults: A Meta-Analysis of Randomized and Non-Randomized Controlled Trials"

_ijerph, 2020, doi:10.3390/ijerph17165643_

Round 1
Reviewer 1 Report
As the authors stated in the literature review, it is important to understand the efficacy of stretching exercises on arterial stiffness in middle-aged and older adults, and the results of meta-analysis of randomized and non-randomized controlled trials could provide significant contribution to the research of this field. However, I think there are several points that may need additional discussion to improve the manuscript:
- The authors stated that they searched through January 2020 databases—what is the starting date?
- I suggest the authors to include more detailed information to explain the screening process of the studies presented in Figure 1. “Records excluded (n=558)”, “Records excluded, with reasons (n= 44)”, “Full-text articles excluded, with reasons (n=17)”, what are the reasons/criteria to exclude these articles?
- Did the authors try to collect any unpublished studies for this meta-analysis? If not, how did the authors try to manage the publication bias?
- The total subjects in the meta-analysis was 213, which is relatively small. The authors may want to comment on this point?
- It seems that some of the studies were female-only, while others were not. Did the authors conduct any analysis to explore the potential gender influence?
Reviewer 2 Report
Very interesting and relevant meta-analysis on arterial stiffness and endothelial function. The analyses support to perform stretching exercises in adults and the elderly, but basically at any age.
The search and compare (stretching + regular life-style/regular life-style) is logical. The relatively small number (213) make the findings of practical relevance (rather than just showing statistical significance).
This paper could be of considerable interest to those working in elderly homes, gyms and in (geriatric) revalidation.
The Tables and Figures are relevant and informative, however consider to select the quintessential Tables and Graphs for the paper in the Journal and provide the a-selected ones as Supplementary Material.
Although I really like the funnel plot (Figure 8), consider to just report the findings of the publication bias results.
The discussion appears somewhat lengthy at first, but as it makes a good read with fine considerations, I would suggest to keep it as is.
I have no comments otherwise.
Reviewer 3 Report
The issue is important, and the study is well done.
Discuss whether the effect and benefit of "Stretching Exercises" are sustained over time.
It will also need to be discussed with the most recent reference: the development of endothelial dysfunction in conduit arteries during aging.
Reviewer 4 Report
The Efficacy of Stretching Exercises on Arterial Stiffness in Middle-aged and Older Adults: A Meta-analysis of Randomized and Non-randomized Controlled Trials
Review
To authors
The authors aimed, through a systematic review with meta-analysis, to investigate whether stretching exercises reduce arterial stiffness in middle-aged and older adults.
The text is well written and organized and the study’s question is clear and coherent along the manuscript. Vascular aging plays a central role in morbidity and mortality of older people. Moreover, the flexibility is one of the components of physical fitness, and, indeed have been receiving some attention in the last years of the literature. Therefore, I am sure that theme is timely and suitable to IJERPH. My main comments are secondary concerns, in order to improve the text, and, I include these aspects here.
First, I think that in introduction the authors can including some epidemiological data on the role of vascular aging in morbidity and mortality in middle-aged and (or) older adults. At very start of introduction (first paragraph), consider including some mechanistic action from aging on vasculature, as cellular and molecular mechanisms of aging, including oxidative stress and mitochondrial dysfunction.
Another concern is about the term stretching, in exercise sciences the stretching can be used in two main modes, as warm-up strategy, and as training method of flexibility. It is a common mistake from non exercise/sport scientists mixing these terms and applications. Please, confirm that the studies used, applied the stretching in which context. If you confirm that was just in a context of warm-up, please consider adding in the final discussion that the study is limited to this condition, without analyze on flexibility aspect of physical fitness. And, even acknowledging the confusion on literature, I recommend using the term stretching as exercises and flexibility as training method/program or capacity/component of physical fitness. For example in the tables you used the term stretching training, sound weird. And, analyzing the articles selected I realize that stretching exercises were not used as warm-up because no further exercise was required.
Reading the articles selected from your data analysis procedures I confirmed that these randomized and non-randomized trials were just applying stretching without combined/concurrent training, but I think you can explicitly indicate that in the methods it.
Even secondary, I think would be very useful if the authors included the outcomes from flexibility adaptations from these interventions, with or without improvements would be interesting to public health professionals. At least, the mean values without analysis should be enough.
Minor points
Consider including keywords different of title.
Lines 71-72 – You inform that three evaluators reviewed the trials. Consider, if so, including the information that two reviewers checked all trials and if have some disparity an third evaluator analyzed the case.
Lines 129-130 – The decision of authors deciding for low risk of bias related to “blinding of participants and personnel” and “blinding of outcome assessment” is correct.
Figure 2 – consider correcting the quality of table’s title similar to remaining table.
Line 148 – std, standardized. Consider including this abbreviation in all figures.
Lines 149-150 – sensitivity analysis-1 ?
Lines 155 – sensitivity analysis 1 ?
Line 172 – std, standardized.
Line 179 – std, standardized.
Figure 7 – the bottom is (slightly) chopped.
